# Gender and Age Differences in Outcomes after Mild Traumatic Brain Injury

**DOI:** 10.3390/jcm12154883

**Published:** 2023-07-25

**Authors:** Sophia Wågberg, Britt-Marie Stålnacke, Beatrice M. Magnusson

**Affiliations:** 1Department of Surgery and Perioperative Sciences, Anesthesiology and Intensive Care Medicine, Umeå University, 901 87 Umea, Sweden; sophia.wagberg@hotmail.com; 2Department of Community Medicine and Rehabilitation, Rehabilitation Medicine, Umeå University, 901 87 Umea, Sweden; britt-marie.stalnacke@umu.se

**Keywords:** traumatic brain injury, post-concussion syndrome, disability, RPQ, GOSE

## Abstract

Many people who suffer traumatic brain injury (TBI) have long-term residual symptoms. This study evaluates post-TBI symptoms and disabilities seven to eight years after mild TBI (mTBI), with specific aims to evaluate gender and age differences, and whether repeated TBI leads to the deterioration of symptoms and function. Telephone interviews with 595 patients were conducted using the Rivermead Post-Concussion Symptoms Questionnaire (RPQ) to assess post-TBI symptoms, and the Glasgow Outcome Scale Extended (GOSE) was used to assess disability. Thirty-four percent reported post-concussion symptoms (40% of females and 29% of males). The symptom burden was higher in women than in men, and higher in patients with repeated TBI. The distribution of symptoms was similar for women and men. Women reported a significantly higher level of disability on GOSE; 31% had not returned fully to daily life, compared with 17% of men (*p* < 0.001), the biggest difference being in the age group of 25–49 years. Patients with repeated mTBI reported significantly lower scores on GOSE; 31% had not returned fully to daily life, compared with 21% of the single-TBI patients (*p* < 0.05). After mild TBI, one of three patients reported at least one post-TBI symptom. Women and individuals with repeated TBI presented a worse GOSE outcome. These findings have implications for clinical practice and research and should be taken into consideration when planning the rehabilitation and follow-up of mTBI patients. This also emphasises the importance of informing patients about post-concussion symptoms and when to seek healthcare.

## 1. Introduction

Traumatic brain injury (TBI) is estimated to affect more than 10 million people globally every year and can have a major impact on the patient’s life [1,2,3]. In the USA, the incidence is 180–250 cases per 100,000 inhabitants per year [4], while in Sweden, the incidence is around 170 cases per 100,000 inhabitants per year [5]. The incidence is highest among young and elderly people, and the most common injury mechanisms are falls and vehicle-related injuries [2]. Men are at greater risk of sustaining TBI, especially in their younger years, while women constitute a larger part of fall-related TBI among the elderly [2,6].

Approximately 90% of treated TBI patients have mild TBI (mTBI) [2,3,4,7]. In the emergency room (ER), the TBI patient’s level of consciousness is examined and rated according to the Glasgow Coma Scale (GCS) score based on eye opening, verbal response, and motoric reaction [8]. A mild TBI corresponds to a GCS of 13–15.

The majority of mTBI patients recover within a couple of months, but as many as 15–50% continue to report symptoms one year after the injury [7,9]. Biological, psychological, and social factors all play a role in the development of post-concussion symptoms. Risk factors for developing post-concussion symptoms are premorbid psychopathology, depression, anxiety, and the patient’s negative perception and expectation of the injury [10]. The Rivermead Post-Concussion Symptoms Questionnaire (RPQ) has been developed [11] to measure post-concussion symptoms after mTBI. Women report a higher symptom burden with a higher RPQ score compared with men, and over a third of patients, regardless of gender, report post-concussion symptoms several years after the injury [1]. The functional status of patients suffering from mTBI is also affected, with them reporting changes in their previous work tasks or a complete job change due to the injury [1,3]. This outcome has been found to differ in severity between the genders [6]. Some studies suggest that a larger difference between genders in disability exists in certain age groups [6,12]. However, a study by Starkley et al. did not find any evidence of poorer outcomes for women in any specific age group, eight years post-mTBI [1]. Repeated mTBI has been shown to result in worse symptom burden and disability [1,13]. Recurrent mTBIs are most common in people under 30 years, with men reporting a higher incidence but women experiencing more post-concussion symptoms [1,13].

Thus, previous studies have shown different results regarding post-mTBI consequences for men, women, and different age groups. Currently, there are only a few studies that follow these patients for a long time, which leaves unanswered questions regarding how long after the injury the patients suffer from post-concussion symptoms. Therefore, the overall aim of this study was to evaluate concussion symptoms and disabilities after mTBI. The specific aims were to evaluate gender and age differences and the relationship between them, and to evaluate whether repeated mTBI leads to the deterioration of symptoms and function.

## 2. Materials and Methods

### 2.1. Participants and Study Design

This study was a single-centre prospective observational cohort study where 1119 patients, aged 25–60 years at the time of the injury, who came to the ER at Umeå University Hospital with an mTBI (GCS 13–15) were included. The age limit was chosen primarily to capture patients of a working age and reduce the bias of memory loss in the oldest patients. Of these, 243 were excluded due to their lack of contact details, blocked records, or because they were no longer alive. Out of the 876 patients, 595 (68%) responded and agreed to participate. Patient data were collected from a database containing mTBI patients who sought healthcare during 2015–2016. Database variables relating to the patient’s TBI included age, sex, arrival status (GCS), injury mechanism, injury location, computed tomography (CT), CT pathology, and admission to the hospital. In the injury location category “other”, the two most common locations were “cross track” and “outside”. For some patients, the injury location could not be found in the patient records and was therefore set as unknown.

### 2.2. Measures

Two questionnaires have been used in this study: the Rivermead Post-Concussion Symptoms Questionnaire (RPQ) to examine post-concussion symptoms, and the Glasgow Outcome Scale Extended (GOSE) to examine functional status.

#### 2.2.1. Symptoms

The RPQ includes 16 items divided into four categories: somatic, cognitive, emotional, and visual symptoms [11]. The symptoms include a headache, dizziness, nausea, noise sensitivity, sleep disturbance, fatigue, being irritable, feeling depressed, feeling frustrated, forgetfulness, poor concentration, taking longer to think, blurred vision, light sensitivity, double vision, and restlessness. Each item was graded on a scale of 0–4, with 0 points meaning no experience of the symptom, 1 point given if the symptom was no greater than before the injury, 2 points given if it was a mild problem, 3 points given if it was a moderate problem, and 4 points given if it was a severe symptom, making 64 points the total highest score possible. In this study, a symptom that was no worse than before the study (given one point in RPQ) was given zero points since we wanted to highlight symptoms that were a result of mTBI.

#### 2.2.2. Disability

The second questionnaire, GOSE, is a disability scale rating from 1 to 8 points. To assess the recovery, the post-discharge structured interview for GOSE was used [14]. The interview examines the ability to function at home, and when shopping, travelling, working, and participating in social activities. Based on these questions, the patients were placed in one of the eight categories which correspond to the scores of 1–8 mentioned above: death, vegetative state, lower or upper severe disability (SD−/+), lower or upper mild disability (MD−/+), and lower and upper good recovery (GR−/+).

### 2.3. Data Collection

The interviews were conducted by two people, one of whom was involved in writing this article. Telephone numbers were collected from patient records if available and otherwise via search engines. After answering the call, the patient was informed about the study and its purpose and asked to give their consent to participate. All interviews followed a script with questions containing RPQ, GOSE, and three additional questions: (1) whether they experienced repeated TBI, regardless of grade, and if so, when, (2) whether they sought healthcare for TBI-related problems, and (3) whether they changed their occupation due to the injury. The phone calls were made during office hours unless someone asked to be called back at another time. In the event of non-response after three attempts, they were excluded.

### 2.4. Statistical Analysis

Age was specified as the patient’s age at the time of injury; hence, the patient is now 7–8 years older. The participants were divided into four age groups (25–29, 30–39, 40–49, and 50–60 years). Only repeated mTBIs after the initial injury in the year of 2015–2016 were counted and divided into three groups: (1) only one mTBI, (2) one to three repeated mTBIs, and (3) more than four repeated mTBIs.

GOSE and the distribution between no, mild, moderate, and severe symptoms were analysed using the chi-square test, and a *p* < 0.05 was regarded as statistically significant. RPQ mean scores were analysed using descriptive statistics for the mean value and a 95% confidence interval (CI). Non-overlapping 95% CI equals *p* < 0.05 and was regarded as statistically significant. Error bars showing 95% CI were included in the figures. Statistically assured results with *p* < 0.05, <0.01, or <0.001, respectively, were marked with asterisks in the figures. Statistical analyses were made with Microsoft Excel (Mac, v.16.72).

### 2.5. Ethics

This study has ethical approval for research on human subjects, Dnr: 2019-05337. All participants and medical records in the study have been handled in accordance with privacy laws, including the General Data Protection Regulation (GDPR).

## 3. Results

### 3.1. Overall Cohort

Table 1 shows the demographic and injury characteristics of the subgroup. The majority of patients were assessed as GCS 15 (97%). A computer tomography (CT) scan was performed in 54% of the cases. Of these, 5.6% were abnormal. Slightly more men than women were admitted for hospital care, 11% and 7.9%, respectively. The most common injury mechanisms were falls (47%), followed by bicycle accidents (15%) and other road traffic injuries (12%). Public place and street/traffic were the two most common injury locations, each with 26%. Notably, the proportion of men injured at work was more than double the proportion of women (8.8% vs. 3.0%). Another injury location with a large gender difference was in the horse stable, with 11% for women and 0.3% for men.

### 3.2. Symptoms

A total of 34% (*n* = 202) of the participants reported one or more of the 16 symptoms in the RPQ. Among these, 40% were women (*n* = 107) and 29% were men (*n* = 95). Figure 1 shows a statistically higher symptom burden for women in the somatic, cognitive, and visual categories of RPQ. The distribution between mild, moderate, and severe symptoms are shown in Figure 2. Approximately 8% of women reported moderate or severe somatic symptoms, compared with 3.9% of men. In the cognitive category, 11% of women reported moderate or severe symptoms, compared with 4.9% of men. The distribution of symptoms was similar between the genders (Figure 3). However, women reported a higher symptom score for each RPQ category (Table 2). The four most common symptoms for both genders were noise sensitivity (17%), forgetfulness (17%), fatigue (16%), and headaches (16%).

As shown in Figure 4, there was no age group where men presented higher mean scores than women. Women between 50 and 60 years showed the lowest symptom burden compared with women in other age groups, and this was the age group with the highest mean score for men. The age group of 50–60 years was therefore the group where men and women had the most even mean score.

Patients with additional mTBI reported significantly higher mean RPQ scores than patients with only the initial mTBI, as seen in Figure 5 (7.0 compared with 3.8, SD 1.96, and 0.73, respectively, *p* < 0.05). The patients with repeated mTBIs reported between one and eight repeated TBIs. One patient reported 20 repeated mTBIs. A statistically significant RPQ score difference was also seen between only one mTBI and 1–3 repeated TBIs (7.1 compared with 3.8, SD 2.05, and 0.73, respectively, *p* < 0.05). No significant difference was seen between 1–3 repeated mTBIs and more than four mTBIs (*n* = 6).

### 3.3. Disability

Ninety-two percent of the mTBI patients showed good recovery, of which 77% showed upper good recovery and 15% showed lower good recovery. However, a larger proportion of women reported a lower functioning in daily life compared with men (Figure 6). Upper good recovery reached 69% for women and 83% for men (*p* < 0.001).

The youngest age groups (25–29 and 30–39) had the best recovery, with 95% in the two highest recovery groups (GR+/−), compared with 90% in the oldest age groups (40–49 and 50–60, *p* < 0.05). No significant differences were seen in the GOSE categories for the different age groups (Figure 7). A significantly lower proportion of women than men reached upper good recovery in the three youngest age groups (29–30, 30–39, 40–49, *p* < 0.05). For the age groups 30–39 and 40–49 years, 17% more men than women reached upper good recovery (Figure 8).

Better recovery was seen for patients with one TBI compared with those who had repeated TBIs (Figure 9). Upper good recovery was reached by 79% of patients with a single TBI, compared with 69% of those with repeated TBIs (*p* < 0.05). Twice as many with repeated TBIs were located in the upper moderate disability group (MD+), compared with those with one TBI (*p* < 0.05).

## 4. Discussion

This study analysed residual symptoms and disability after mild TBI with the aim of clarifying whether gender, age, or repeated TBI had an impact on the outcome after mTBI. Our results demonstrate that as many as one of three patients still reported post-TBI symptoms 7–8 years after the initial injury. A higher percentage of women than men reported symptoms, especially somatic, cognitive, and visual symptoms (*p* < 0.05). Women also reported symptoms of greater severity, the difference being greatest for the somatic, cognitive, and visual symptoms.

In a study made by Starkey et al., approximately 30% of patients met the cut-off for post-concussion syndrome (PCS) after eight years, according to DSM-IV-criteria [1]. These criteria meant that the patient had to report at least moderate symptoms for a minimum of three symptoms. However, it should be taken into consideration that the DSM-IV-criteria were not applied to our study population and if they had been, they would have lowered the proportion of participants counted as affected in our study. On the other hand, in Starkey’s study, 36% were reported to be affected by their injury regardless of any minimum point criterion. This number corresponds well with our results, which showed that 34% were affected by post-concussion symptoms seven to eight years after the initial mTBI.

In line with previous studies, we found that women reported more symptoms in all RPQ categories and suffered from worse functional impairments, according to GOSE. Mikolic et al. found that the differences in recovery between men and women were more pronounced in ages under 45 and over 65 [6]. Starkey et al. could not find any significant differences regarding symptoms in any specific age group [1]. We could not study the differences in the symptoms between the genders and age groups due to the lack of significance. However, when studying disability, we found that there was a significantly larger proportion of women than men who had lower functioning in all age groups except 50–60 years. For the other three age groups, the difference between men and women was very similar, between 14–17%. The U-shaped trend of women recovery, with the highest recovery in the lowest and highest age groups, is in line with the age-related pattern of women diagnosed with exhaustion syndrome, where women are over-represented [15]. This may be due to the larger responsibility that Swedish women take regarding their children and household [15,16]. Patients in the 50–60 age group are now between 57 and 68 years old, which means that any children they had will have moved out, the patients are close to retirement or recently retired, and the patients have more time for recovery.

Previous studies have shown more post-concussion symptoms for people with repeated mTBIs [13,17]. Theadom et al. showed that repeated TBIs significantly increased the risk of PCS one year after the initial index injury [13]. However, they could not find any difference in disability between single and repeated TBIs. In our study, we found that patients with repeated TBIs had a significantly higher mean RPQ score, and a higher proportion had not fully recovered to daily life according to GOSE. When comparing the mean scores for no repeated injuries and a few additional mTBIs (*n* = 1–3), we could show a significant difference (*p* < 0.05). Since no significant difference was found in the group with more than four repeated mTBIs, it was difficult to draw conclusions about how the increase in the number of repeated mTBIs affected the symptom burden and functional disability of the patients.

### Strengths and Limitations

One strength of our study was the high participation rate (68%), as it included many participants. The fact that patients were contacted by telephone and were able to perform the interview directly may have contributed to a higher response rate than if a survey had been sent out by post or online. Another strength is that all the interviews were conducted by only two people who were following a manuscript, which reduces the risk of different interpretations of the answers. The patients presented a wide spread of injury mechanisms and locations, making the study widely applicable compared with a study on, for example, only athletes.

This study does have a few limitations. Firstly, there was no healthy control group, which makes it difficult to draw conclusions based on the overall percentage of reported symptoms. The symptoms included are experienced by many people, regardless of brain injury. Some of the symptoms are more common in women, which may affect the outcome. For example, migraines have been reported to be three times more common for women than men [18]. However, it is interesting that the distribution between the genders was so similar when the burden of symptom differs significantly. This may indicate that the symptoms were post-concussive and cannot be explained only by gender differences in individual symptoms.

Another limitation is the length of time that had passed since the injury, which may lead to a recall bias. The patient was asked whether the symptoms were worse now than before the injury. It can be difficult to assess one’s state of health after such a long time, especially for elderly patients with some kind of memory problem. However, the long follow-up time was one of our aims and provides an interesting aspect regarding the possibility of investigating the long-term effects of TBI, for which only a few studies have been performed previously. It should be noted in this context that post-concussion symptoms are not exclusively encountered in patients with mTBI; they are also frequently reported in, for example, patients with chronic pain and patients who have suffered whiplash injuries [19]. In addition, since we had no information regarding treatment or rehabilitation interventions, we cannot rule out the potential effects of such aspects.

## 5. Conclusions

We are elaborating on previous findings on gender and age differences after single and repeated mTBI. We have shown that women have poorer outcomes after mTBI, as is the case for patients with repeated TBIs. These results have implications for clinical practice and research and should be taken into consideration when planning the rehabilitation and follow-up of mTBI patients. This also emphasises the importance of informing patients about post-concussion symptoms and when to seek healthcare, as it has been shown that a large percentage of mTBI patients do not receive adequate instructions or follow-up [20]. Our study confirms the results of previous studies, namely that repeated TBI can have long-lasting and serious consequences [13]. Therefore, the important preventive work being done, in sports, for example, must continue. Further studies are needed since there are gaps in the follow-up care for patients with mTBI after hospital discharge. These studies should preferably include several different hospitals to contain a wide range of patients.

## Figures and Tables

**Figure 1 jcm-12-04883-f001:**
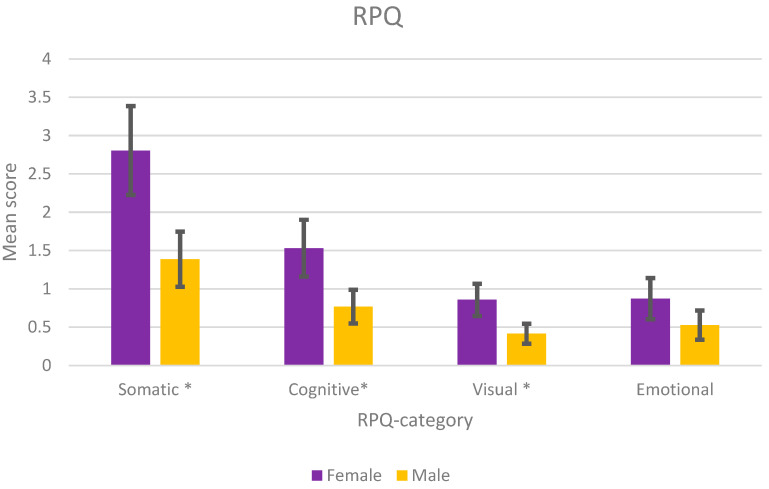
Mean score for males and females in each of the RPQ categories. * *p* < 0.05.

**Figure 2 jcm-12-04883-f002:**
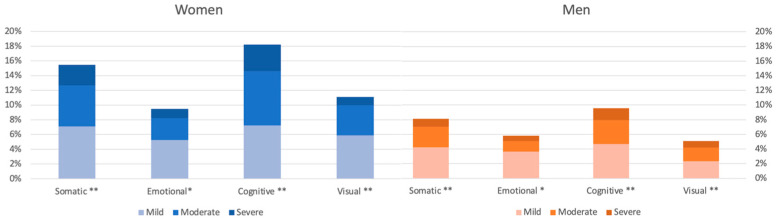
The percentage of patients with RPQ-symptoms, divided by gender and severity. * *p* < 0.05 ** *p* < 0.001.

**Figure 3 jcm-12-04883-f003:**
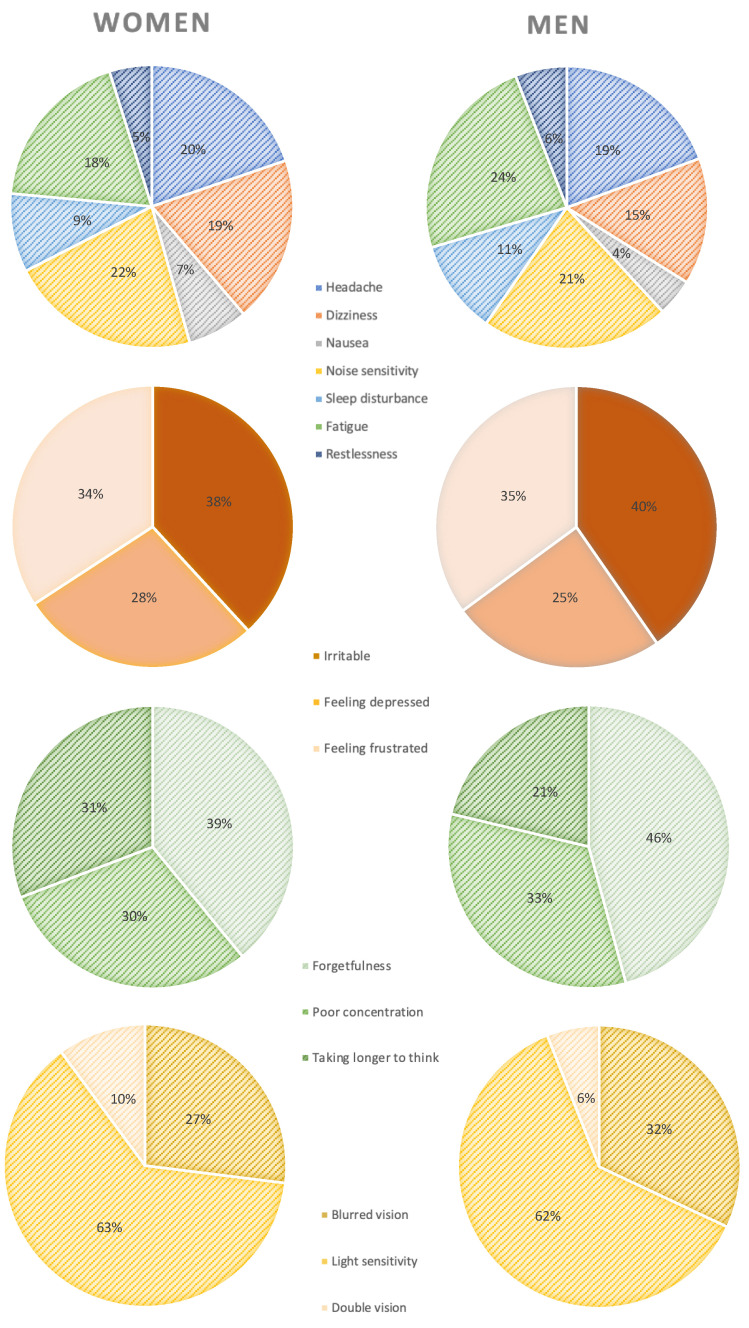
The percentage of reported somatic, emotional, cognitive, and visual RPQ symptoms divided by gender.

**Figure 4 jcm-12-04883-f004:**
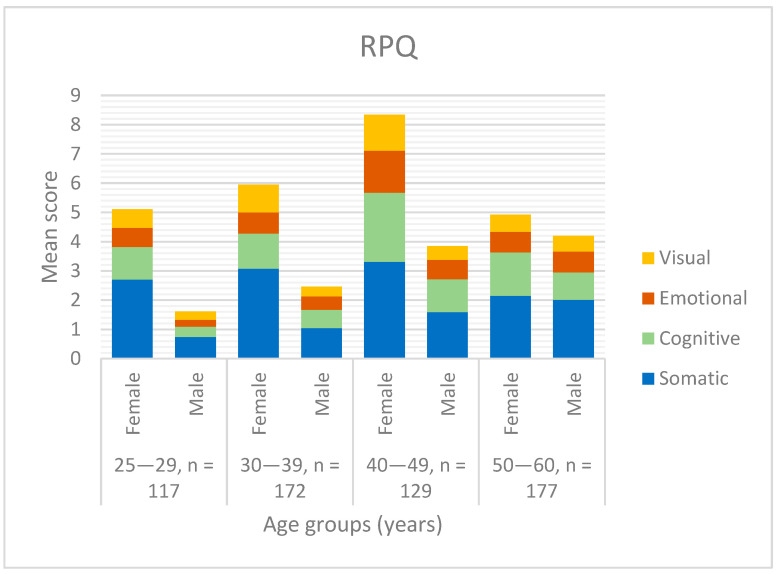
Mean RPQ score for the different gender and age groups.

**Figure 5 jcm-12-04883-f005:**
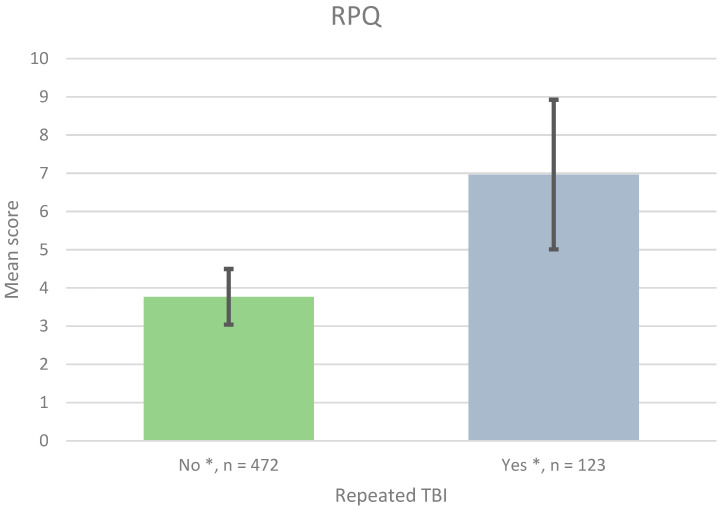
Mean RPQ score for patients with none (No), respectively, additional TBI (≥1) after the initial injury (Yes). * *p* < 0.05.

**Figure 6 jcm-12-04883-f006:**
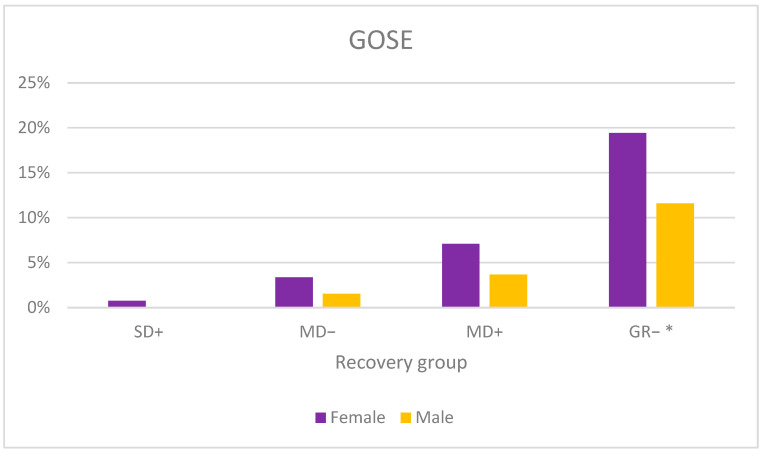
The percentage of GOSE distribution divided by gender. GR−: Lower good recovery, MD+: upper moderate disability, MD−: lower moderate disability, SD+: upper severe disability. Upper good recovery is not presented. * *p* < 0.01.

**Figure 7 jcm-12-04883-f007:**
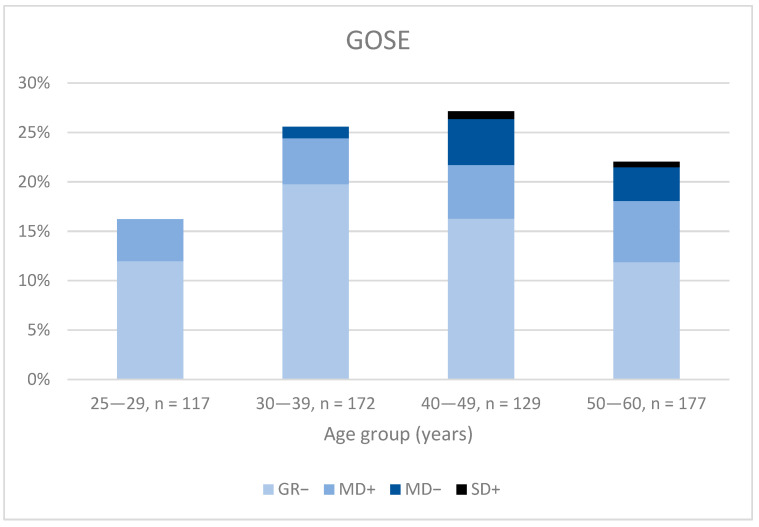
The percentage of GOSE distribution divided by age group (years GR−: Lower good recovery, MD+: upper moderate disability, MD−: lower moderate disability, SD+: upper severe disability. Upper good recovery is not presented.

**Figure 8 jcm-12-04883-f008:**
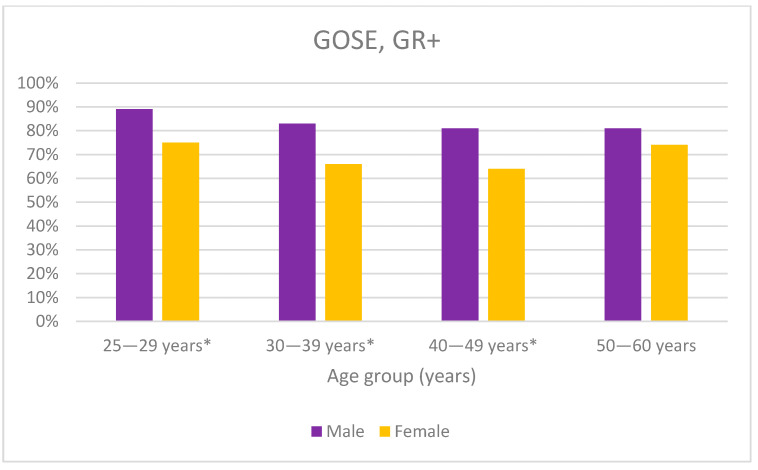
The percentage of men and women in GR+, upper good recovery, in each age group. * *p* < 0.05.

**Figure 9 jcm-12-04883-f009:**
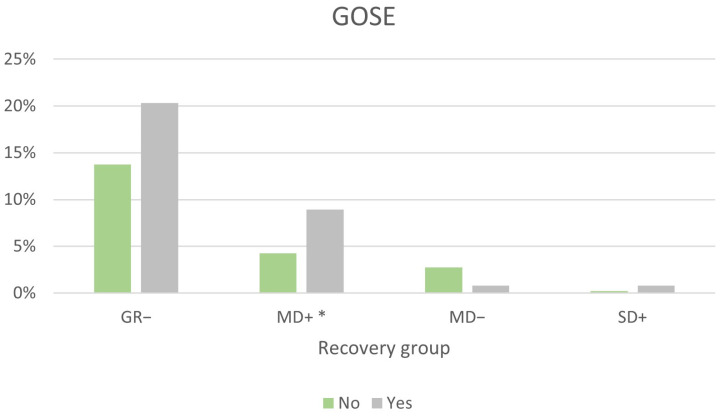
The percentage of GOSE distribution divided by none (No), respectively, repeated (≥1) TBI (Yes) after the initial injury. GR−: Lower good recovery, MD+: upper moderate disability, MD−: lower moderate disability, SD+: upper severe disability. Upper good recovery is not presented. * *p* < 0.05.

**Table 1 jcm-12-04883-t001:** Demographic and injury characteristics for the included patients. RTA = road traffic accidents. CT = computer tomography. GCS = Glasgow Coma Scale. Statistically significant results are marked with bold.

Category	Subcategory	Gender						*p*-Value
		Male		Female		Total		
		*n*	%	*n*	%	*n*	%	
**Patients**		328		267		595	100	
**Injury mechanism**	Assault	29	8.8	9	3.4	38	6.4	**<0.01**
	Fall height	21	6.4	31	12	52	8.7	**<0.05**
	Fall standing	111	34	115	43	226	38	**<0.05**
	Fall wheelchair	1	0.3	0	0	1	0.2	0.37
	Other	47	14	24	9.0	71	12	**<0.05**
	RTA	43	13	30	11	73	12	0.49
	RTA cyclist	52	16	35	13	87	15	0.35
	Sport	24	7.3	23	8.6	47	7.9	0.56
**Injury location**	Home	66	20	67	25	133	22	0.15
	Work	29	8.8	8	3.0	37	6.2	**<0.01**
	Other	18	5.5	9	3.4	27	4.5	0.22
	Horse stable	1	0.3	30	11	31	5.2	**<0.001**
	Public place	88	27	68	25	156	26	0.71
	Sport field	21	6.4	10	3.7	31	5.2	0.15
	Street/Traffic	86	26	67	25	153	26	0.75
	Unknown	19	5.8	8	3.0	27	4.5	0.10
**CT-scan**	Performed	188	57	135	51	323	54	0.10
	Abnormal	14	7.4	4	3.0	18	5.6	0.05
**Admission**	Yes	35	11	21	7.9	56	9.4	0.24
**GCS**	15	316	96	262	98	578	97	0.19
	14	8	2.4	4	1.5	2	1	0.42
	13	4	1.2	1	0.4	5	0.8	0.26

**Table 2 jcm-12-04883-t002:** The number and percentage of reported RPQ symptoms. Statistically significant results are marked with bold.

RPQ Subgroup	RPQ Symptoms	Male		Female		Total		*p*-Value
Patients		*n* = 328	%	*n* = 267	%	*n* = 595	%	
**Somatic**	Headaches	36	11	57	21	93	15.6	**<0.001**
	Dizziness	27	8.2	55	20.6	82	13.8	**<0.001**
	Nausea	8	2.4	20	7.5	28	4.7	**<0.01**
	Noise sensitivity	40	12.2	63	23.6	103	17.3	**<0.001**
	Sleep disturbance	20	6.1	26	9.7	46	7.7	0.098
	Fatigue	44	13.4	54	20.2	98	16.5	**<0.05**
	Restlessness	11	3.4	14	5.2	25	4.2	0.25
**Emotional**	Irritable	23	7.0	29	10.9	52	8.7	0.098
	Feeling depressed	14	4.3	21	7.9	35	5.9	0.064
	Feeling frustrated	20	6.1	26	9.7	46	7.8	0.098
**Cognitive**	Forgetfulness	43	13	57	21.3	100	16.8	**<0.001**
	Poor concentration	31	9.5	44	16.5	75	12.6	**<0.05**
	Taking longer to think	20	6.1	45	16.9	65	10.9	**<0.001**
**Visual**	Blurred vision	16	4.9	24	9.0	40	6.7	**<0.05**
	Light sensitivity	31	9.5	56	21.0	87	14.6	**<0.001**
	Double vision	3	0.9	9	3.4	12	2.0	**<0.05**

## Data Availability

The data that support the findings of this study can be made available on request from the corresponding author. The data are not publicly available due to ethical restrictions.

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
