# Peer review of "Gender and Age Differences in Outcomes after Mild Traumatic Brain Injury"

_jcm, 2023, doi:10.3390/jcm12154883_

Round 1

Reviewer 1 Report

The manuscript deals with the evaluation of long-term effects in gender and age differences among patients who suffered of traumatic brain injuries. Telephone interviwes involving 595 participants were conducted. Two questionnaires were used, namely (a) The Rivermead Concussion Symptom Questionnaire, and (b) The Glasgow Outcome Scale Extended. Results evidenced that the symptom burden was higher in women as well as the level of disability while the distribution og symptoms was similar between men and women. Different long-term outcomes were additionally recorded. 

Although I found the paper interesting and appropriate for the Journal, I feel that relevant issues should be saddresed in a suitable revision. My points are listed below. 

1. The Introduction is incomplete and should be improved. I wonder whether the authors considered both assessment and recovery/rehabilitation features in their theoretical framework and whether the disorders of consciousness (i.e., vegetative state, minimally conscious state or emerging from it) were included as potential variables in the recovery process and in the effects of TBI towards the long-term outcomes. Were different approaches/strategies referenced? I think that the point is mandatory to provide an exhaustive framework and support a strong rationale for the current study. 

2. In the Method section, it was unclear to me how the participants were recruited. Eligibility criteria (including and excluding criteria) should be detailed. 

3. Each figure should be an X label and a Y Label, a progressive numeral and a caption. If the authors reger to a percentage the ranging scale should be 0-100. 

4. The Discussion should include the implications of the findings for both research and practice. Relevant citations should be added. 

5. The conclusion should be enhanced. Future research perspectives in this specific framework should be included. 

Author Response

Point to point response

By Sophia Wågberg

We are grateful for the comments presented by the reviewers, which were very valuable. Here below you will find how we dealt with the criticism and suggestions and how the manuscript is revised. The changed text is highlighted with the change tracking function. Please see the attachment for the revised manuscript. 

Reviewer 1

Answer from Authors

1.The Introduction is incomplete and should be improved. I wonder whether the authors considered both assessment and recovery/rehabilitation features in their theoretical framework and whether the disorders of consciousness (i.e., vegetative state, minimally conscious state or emerging from it) were included as potential variables in the recovery process and in the effects of TBI towards the long-term outcomes. Were different approaches/strategies referenced? I think that the point is mandatory to provide an exhaustive framework and support a strong rationale for the current study. 

In the revised manuscript, Introduction, we have clarified that our study only included patients with mild TBI, GCS 13-15 to fulfil the requirements.

Moreover, since the study population was patients that were assessed at the ER at Umeå University Hospital, Sweden, we do not know if they received any rehabilitation intervention. Thus, we could not consider treatment and rehabilitation as potential variables in the recovery process.

In Discussion, Limitations the following sentence has therefore been added on page 11, line 322-323: “In addition, we had no information regarding treatment or rehabilitation interventions, therefore we cannot rule out the potential effects of such aspects”. 

2.In the Method section, it was unclear to me how the participants were recruited. Eligibility criteria (including and excluding criteria) should be detailed. 

In the revised manuscript the recruitment procedure for the patients has been clarified on page 2, in section: participants and study design to fulfil your requirements. Inclusion and exclusion criteria have now been added in more details.

3.Each figure should be an X label and a Y Label, a progressive numeral, and a caption. If the authors reger to a percentage the ranging scale should be 0-100. 

Y and X-labels have been added. In order to keep the graphs easy to overlook, the y-axis does not range between 0-100% in figures 2, 6, 7 and 9, respectively. Each figure has a caption with a progressive numeral and a legend.

4. The Discussion should include the implications of the findings for both research and practice. Relevant citations should be added. 

The Discussion and Conclusion now include implications of the results for clinical practice and further studies to fulfil your requirements, see pages 11-12.

5. The conclusion should be enhanced. Future research perspectives in this specific framework should be included. 

See 4 above.

Reviewer 2 Report

Authors from Sweden investigated post-traumatic brain injury (TBI) symptoms and disabilities seven to eight years after mild TBI to evaluate gender and age differences. Telephone interviews with 595 patients were conducted. The symptom burden was higher in women than in men, and higher in patients with repeated TBI.

1.       In abstract the description seem to contain background, methods and results but not a conclusion.

2.       L70 Why only patients aged 25-60 years at the time of injury were included, why no younger and no older?

3.       Table 1- maybe you should insert p-values for comparison between women and men here

4.       Figure 1- also here p-values for comparison between women and men here.

5.       Probably better to show p value for each pair of bars in each Figure

6.       Further limitation is that this is a single center study

7.       Further limitation is that some patients may have memory problems biasing their responses

Author Response

Point to point response

By Sophia Wågberg

We are grateful for the comments presented by the reviewers, which were very valuable. Here below you will find how we dealt with the criticism and suggestions and how the manuscript is revised. The changed text is highlighted with the change tracking function. Please see the attachment for the revised manuscript. 

Reviewer 2.

Answer from Authors

1.   In abstract the description seem to contain background, methods and results but not a conclusion.

In the revised manuscript we have now added the following conclusion on page 1, line 23-26: “These findings have implications for clinical practice and research and should be taken into consideration when planning rehabilitation and follow-up of mTBI patients”.

2. L70 Why only patients aged 25-60 years at the time of injury were included, why no younger and no older?

This has been clarified in the Material and methods section where the following sentence has been added on page 2, line 79-82: “The age limit was chosen primarily to capture patients of working age and reduce the bias of memory loss in the oldest patients”.

A sentence regarding this has also been added on page 11, line 315-316 in the Limitation section; “Especially for elderly patients with some kind of memory problem”.

t.

3. Table 1- maybe you should insert p-values for comparison between women and men here

P-values has been added as requested and are now inserted in Table 1.

4. Figure 1- also here p-values for comparison between women and men here.

In figure 1, p-value <0.05 is marked with an asterisk to fulfil your requirement.

5.  Probably better to show p value for each pair of bars in each Figure

The bars are made with 95% confidence intervals which equals p-value <0.05, and therefore there is not an individual p-value for each set of bars. To fulfil your requirement a new sentence has been added to the Materials & Method section to clarify this issue. Page 3, line 148-149: “Non-overlapping 95% CI equals p<0.05 and was regarded as statistically significant”.

6.Further limitation is that this is a single center study

A sentence regarding this has been added to the conclusion on page 12, line 342-344: “These studies should preferably include several different hospitals to contain a wide range of patients.”

7. Further limitation is that some patients may have memory problems biasing their responses

The fulfil the requirement regarding limitation due to memory problems a new sentence has been added on page 11, line 316; “…especially for elderly patients with some kind of memory problem”.

Round 2

Reviewer 1 Report

I think that my issues raised in the original review have now been addressed.